# Evaluating probabilistic dengue risk forecasts from a prototype early warning system for Brazil

Rachel Lowe[1]*, Caio AS Coelho[2], Christovam Barcellos[3], Marilia Sá Carvalho[3], Rafael De Castro Catão[1,4], Giovanini E Coelho[5], Walter Massa Ramalho[6], Trevor C Bailey[7], David B Stephenson[7], Xavier Rodó[1,8]

[1]Climate Dynamics and Impacts Unit, Institut Català de Ciències del Clima, Barcelona, Spain; [2]Centro de Previsão de Tempo e Estudos Climáticos, Instituto Nacional de Pesquisas Espaciais, Cachoeira Paulista, Brazil; [3]Fundação Oswaldo Cruz, Rio de Janeiro, Brazil; [4]Faculdade de Ciências e Tecnologia, Universidade Estadual Paulista, Presidente Prudente, Brazil; [5]Coordenação Geral do Programa Nacional de Controle da Dengue, Ministério da Saúde, Brasília, Brazil; [6]Faculdade de Ceilândia, Universidade de Brasília, Brasília, Brazil; [7]Exeter Climate Systems, College of Engineering, Mathematics and Physical Sciences, University of Exeter, Exeter, United Kingdom; [8]Institució Catalana de Recerca i Estudis Avançats, Barcelona, Spain

**Abstract** Recently, a prototype dengue early warning system was developed to produce probabilistic forecasts of dengue risk three months ahead of the 2014 World Cup in Brazil. Here, we evaluate the categorical dengue forecasts across all microregions in Brazil, using dengue cases reported in June 2014 to validate the model. We also compare the forecast model framework to a null model, based on seasonal averages of previously observed dengue incidence. When considering the ability of the two models to predict high dengue risk across Brazil, the forecast model produced more hits and fewer missed events than the null model, with a hit rate of 57% for the forecast model compared to 33% for the null model. This early warning model framework may be useful to public health services, not only ahead of mass gatherings, but also before the peak dengue season each year, to control potentially explosive dengue epidemics.

*For correspondence: rachel.lowe@ic3.cat

**Competing interests:** The authors declare that no competing interests exist.

## Introduction

Dengue is an arboviral infection of major international public health concern (*Guzman and Harris, 2015*). Dengue is endemic in more than 100 countries in the tropics and sub-tropics, with Brazil reporting more cases than any other country in the world (*Bhatt et al., 2013*; *Teixeira et al., 2009*). Dengue is caused by four distinct dengue virus serotypes (DENV 1–4), which are transmitted to humans by *Aedes* mosquitoes (*Guzman and Harris, 2015*). The distribution of both *Ae. aegypti* and *Ae. albopictus* is widespread across Brazil, with *Ae. aegypti* found predominantly in urban settings, breeding in artificial containers, and *Ae. albopictus* more commonly found in rural and peri-urban settings (*Kraemer et al., 2015*). However, dengue incidence is unequally distributed in Brazil, with higher and sustainable incidence along the Atlantic coast and in the central region. Temperature and rainfall regimes seem to control the magnitude and seasonality of dengue transmission (*Campbell et al., 2015*). Large outbreaks are typically observed after rainy and warm periods, at the end of summer, particularly in densely populated urban areas (*Teixeira et al., 2009*). The presence

**eLife digest** Dengue is a viral infection spread by mosquitoes and is widespread in tropical and sub-tropical regions. Dengue epidemics in Brazil often occur without warning, and can overwhelm the public health services. Forecasts of seasonal climates combined with early data from a dengue surveillance system could help public health services anticipate dengue outbreaks several months in advance. However, this information has not been previously exploited to predict dengue epidemics in a practical real-life framework.

Recently, a group of researchers developed a prototype of a dengue early warning system based on 13 years worth of data, and used it to predict the risk of dengue three months ahead of the 2014 FIFA World Cup in Brazil. Now Lowe et al. – including most of the researchers involved in the earlier work – have evaluated the prototype against the actual reported cases of dengue during the event. Brazil is divided into over 550 'microregions', and the forecasts correctly predicted high risk of dengue for 57% of the microregions reporting high levels of dengue during the games. Forecasts based on seasonal dengue averages would have only detected high risk in 33% of these microregions. The forecasts also correctly predicted the dengue risk level in seven out of the twelve cities where the World Cup games were hosted. However, the prototype failed to predict the high risk in both São Paulo and Brasília. Lowe et al. speculate that this may have been due to changes in how water was stored in these cities (standing water is a breeding site for mosquitoes) and the circulation of a new strain of the dengue virus.

The implementation of seasonal climate forecasts and early reports of dengue cases into an early warning system is now a priority for public health authorities. This action is likely to help them to prepare for and minimize epidemics of dengue and other diseases that are spread by mosquitoes, such as chikungunya and Zika virus.

and abundance of dengue mosquitoes is a necessary but not sufficient condition for dengue transmission and the occurrence of large outbreaks. Besides vector infestation, an important factor regulating transmission is the introduction of new serotypes of virus in areas with a high susceptible population. This may be facilitated by the increasing international and internal mobility across the country. Thus, large and touristic cities are prone to the introduction and maintenance of virus circulation. International mass gathering events have become an important health issue in the recent years (*Abubakar et al., 2012*) as they create the opportunity for the introduction of new pathogens in a susceptible population, as well as exposing visitors to new and unknown local risks (*Matos and Barcellos, 2010*).

Early warning systems, which take into account multiple dengue risk factors, can assist public health authorities to implement timely control measures ahead of imminent dengue outbreaks. Seasonal climate forecasts combined with early dengue surveillance system data provide an opportunity to anticipate dengue outbreaks several months in advance (*Lowe et al., 2014*). To date, several studies have assessed the use of climate information in early warning systems for diseases, such as malaria and Rift Valley fever (*Anyamba et al., 2009*; *Thomson et al., 2006*). The incorporation of climate information for dengue early warning systems has also been explored (*Degallier et al., 2010*; *Lowe et al., 2011*; *Stewart-Ibarra and Lowe, 2013*). However, to our knowledge, real-time climate forecasts have not been previously applied to predict dengue epidemics in a practical real-life framework.

From 12 June to 13 July 2014, Brazil hosted the 2014 Fédération Internationale de Football Association (FIFA) World Cup, a mass gathering of more than 3 million Brazilian and international spectators, travelling between 12 different host cities. Before the event, the potential risk of transmission of several communicable diseases, including dengue fever, was highlighted (*Gallego et al., 2014*; *Wilson and Chen, 2014*). Several research groups published dengue outlooks ahead of the World Cup. Approaches included analysing historical time series distributions of city or state level data (*Hay, 2013*) and mapping of historical averages, while accounting for seasonality and areas of permanent transmission (*Barcellos and Lowe, 2014a*). Some groups formulated deterministic (*Massad et al., 2014*) and statistical (*van Panhuis et al., 2014*) models to estimate the number of

tourists expected to contract dengue fever. Another study (*Lowe et al., 2014*) assessed the potential for dengue epidemics during the tournament by providing probabilistic forecasts of dengue risk for the 553 microregions of Brazil with risk-level warnings issued for the 12 cities where the matches were played. The dengue early warning system, formulated using a Bayesian spatio-temporal model framework (*Lowe et al., 2011*, *2013*), was driven by real-time seasonal climate forecasts for the period March-April-May and the dengue cases reported to the Brazilian Ministry of Health in February 2014. This information was combined to produce a dengue forecast at the start of March 2014. Predicted probability distributions of dengue incidence rates (DIR) were summarised and translated into risk warnings, using a two-tier threshold approach. First, the probability of DIR falling into categories of low, medium and high risk was determined using dengue risk thresholds of 100 and 300 cases per 100,000 inhabitants, defined by the National Dengue Control Programme of the Brazilian Ministry of Health (*Ministério da Saúde, 2008*). Second, probability trigger thresholds were calculated by selecting optimal cut-off values that maximised sensitivity and specificity, for each dengue risk threshold (medium and high). Using criteria related to the probability trigger thresholds, forecast warnings of low, medium or high dengue risk were determined for the 12 microregions hosting World Cup matches (see Materials and methods for further details). The forecasts were produced and made available three months ahead of the event (see *Lowe et al., 2014*). In this article, we provide an evaluation of the forecast model predictions by using the observed dengue incidence rates for June 2014 to assess the ability of the model framework to successfully assign dengue risk warning categories for the host microregions and all microregions in Brazil. We also compare the forecast model framework to a null model, based on seasonal averages of previously observed dengue incidence. We then discuss the challenges and limitations of producing disease risk forecasts in a real-time setting, such as the use of incomplete surveillance data to drive the model, the coarse spatial resolution of the forecasts, the definition of risk and alarm trigger thresholds, the lack of information regarding the (re)introduction of different serotypes or vector control activities, and the difficulties in communicating probabilistic forecasts. Finally, we suggest future model developments and advocate a multi-model approach to dengue prediction in the future.

## Results

### Comparison of probability forecast and observed dengue risk categories

*Table 1* shows the dengue forecasts for June 2014 for the microregions where stadiums were located, issued three months before the World Cup, and published before the event (*Lowe et al., 2014*). For comparison, the observed DIR values are included in the table, along with the observed risk categories, determined using the medium and high dengue risk thresholds. Note that the dengue risk thresholds used by the National Dengue Control Programme are based on yearly dengue incidence rates (*Ministério da Saúde, 2008*). Therefore, we converted the monthly incidence to yearly equivalent incidence to make use of the medium and high dengue risk thresholds at a monthly time scale (see Materials and methods). *Figure 1* shows the corresponding predictive distributions, the posterior predicted mean DIR and upper 95% prediction interval (dashed and dotted lines) and observed DIR (marked with an arrow). The comparison of the second and last columns of *Table 1* reveals that the model correctly predicted dengue risk categories (highlighted in blue) for Fortaleza and Natal (high), Belo Horizonte, Manaus and Salvador (medium) and Curitiba and Porto Alegre (low). In Recife, the predicted category was high, but the observed category was medium. However, for Recife, the mean predicted DIR was almost equal to the observed DIR and the point estimate fell within the medium category (see *Table 1*, *Figure 1*). The definition of the alarm trigger threshold placed this microregion in the high category. This result highlights the difficulties of translating probabilistic information into simpler warnings, based on predefined probability trigger thresholds. The model 'missed' the unprecedented high incidence that was observed in Brasília and São Paulo in June 2014. However, the model predicted a 7% forecast probability of observing high risk in Brasília.

Probabilistic forecasts were generated not only for the twelve host microregions, but for all 553 microregions of Brazil. This gives an idea of how the model framework might contribute towards a nationwide dengue early warning system in the future. *Figure 2* shows a ternary probabilistic

**Table 1.** Dengue risk forecast warnings and corresponding observations for June, 2014 for host microregions. Dengue risk forecast warnings and observed category for June 2014, for the microregions hosting the World Cup tournament. Low risk was defined as fewer than 100 cases per 100,000 inhabitants, medium risk as between 100 and 300 cases per 100,000 inhabitants, and high risk as greater than 300 cases per 100,000 inhabitants. If the probability of low risk was less than 68%, a medium risk forecast warning was issued. If the probability of high risk was concurrently greater than 18%, the forecast warning was upgraded to high risk. The observed DIR value is included. Microregions where the observed DIR fell into the same category as forecast (e.g. the forecast warning category was high and the observed DIR category was high), are shaded.

| Microregion | Forecast warning | Probability ($p_L$, $p_M$, $p_H$) | Observed DIR | Observed category |
|---|---|---|---|---|
| Belo Horizonte | Medium | p(65%, 24%, 11%) | 126 | Medium |
| Brasília | Low | p(73%, 20%, 7%) | 725 | High |
| Cuiabá | Low | p(71%, 22%, 7%) | 168 | Medium |
| Curitiba | Low | p(100%, 0%, 0%) | 4 | Low |
| Fortaleza | High | p(34%, 20%, 46%) | 507 | High |
| Manaus | Medium | p(63%, 25%, 12%) | 110 | Medium |
| Natal | High | p(32%, 20%, 48%) | 780 | High |
| Porto Alegre | Low | p(100%, 0%, 0%) | 1 | Low |
| Recife | High | p(57%, 24%, 19%) | 161 | Medium |
| Salvador | Medium | p(56%, 27%, 17%) | 149 | Medium |
| São Paulo | Low | p(99%, 1%, 0%) | 161 | Medium |
| Rio de Janeiro | Medium | p(62%, 25%, 13%) | 32 | Low |

forecast map (*Jupp et al., 2012*; *Lowe et al., 2014*) and the corresponding observed dengue incidence rate categories (low, medium and high). The model correctly predicted, with high certainty (the greater the colour saturation, the greater the certainty), low dengue risk in South Brazil and large areas of the Amazon. Areas with a higher chance of observing high risk were correctly detected for areas in North East Brazil. Actual dengue incidence rates were higher than expected in Brasília, although the likelihood of observing higher dengue incidence for the surrounding region was relatively greater than observing lower incidence. For some microregions in the state of São Paulo, the model was uncertain of the most likely category (indicated by pale colours). Some of these areas experienced high dengue incidence rates in June 2014.

*Figure 3* shows the probability of DIR falling in the category that was actually observed. The deeper the colour shading, the greater the probability of observing the correct category. This gives an indication of the certainty of the model in predicting correct outcomes. In general, a high degree of certainty in the forecast is found in the south region, parts of the Amazon and many densely populated cities along the eastern coastline. However, as the historical distribution of DIR is not symmetrical, with a greater proportion of the distribution in the low category, compared to the high category (as epidemics can be considered as 'extreme events'), it is interesting to consider each category individually. *Figure 4a–c* show conditional maps of the forecast probability given that low, medium and high DIR was observed, respectively. The grey areas indicate areas where the observed DIR fell in the other two categories and are therefore not considered in each individual map. The probability trigger thresholds defined in *Lowe et al., (2014)* are taken into account to weight the graduated colour bars, ranging from 0% to 100% chance of the observed category. Using the forecasting model, if the probability of low risk were greater than 68%, a low risk warning would have been assigned. If the probability of low risk were less than or equal to 68%, a medium risk warning would have been assigned (giving a medium trigger threshold of 32%). If simultaneously, high risk were greater than 18%, a high risk warning would have been assigned. Therefore, lower probabilities are assigned more weight (represented by colour darkness) in the high category plot than the low category plot. Given that low risk was observed, the model framework would have correctly assigned a low risk warning for 67% of the microregions. Given that high risk was observed, the model framework would have correctly assigned a high risk warning for 57% of the microregions.

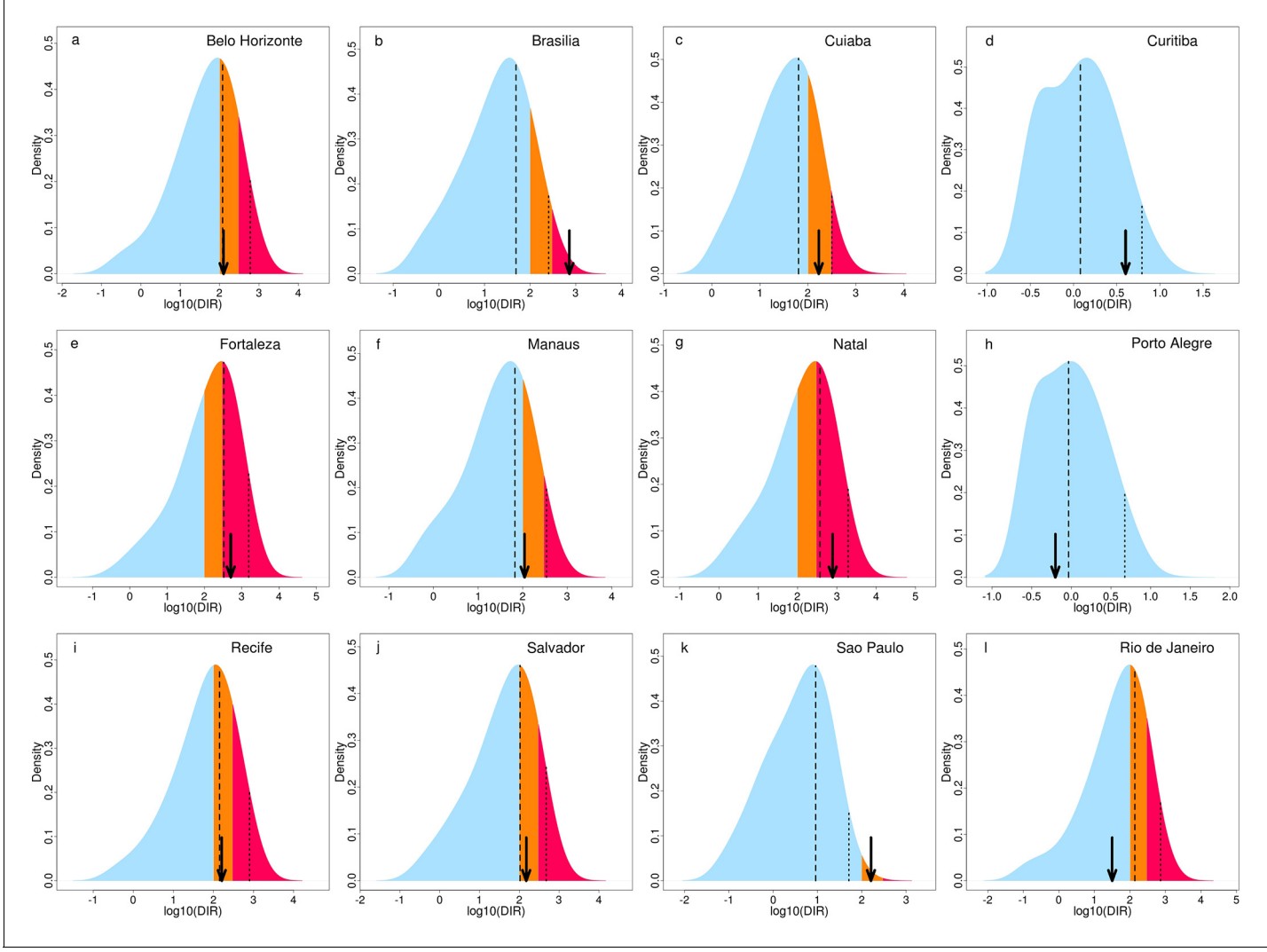

**Figure 1.** Predictive distributions and observed DIR for June 2014 for host microregions. Posterior predictive distributions of dengue incidence rates (DIR) (base-10 logarithmic scale) for June 2014 showing the probability of low risk (blue), medium risk (orange) and high risk (pink) for June 2014, in the microregions hosting the World Cup tournament: (a) Belo Horizonte, (b) Brasília, (c) Cuiabá, (d) Curitiba, (e) Fortaleza, (f) Manaus, (g) Natal, (h) Porto Alegre, (i) Recife, (j) Salvador, (k) São Paulo and (l) Rio de Janeiro. Observed DIR indicated by black arrow. Posterior predictive mean and upper 95% prediction (credible) interval of the distribution indicated by a dashed and dotted line, respectively.

High risk was correctly forecast with considerable certainty in microregions in the north east of Brazil near Fortaleza (see *Figure 4c*). Although the model 'missed' the high risk observed in Brasília, it was able to correctly detect, with a relatively high degree of certainty high risk in surrounding microregions.

## Comparison of forecast model framework to a null model

Useful predictions from a forecasting system are likely to be those that recommend changes from the activities that would otherwise have taken place anyway, which are typically based on the 'normal' dengue season. Beyond that, predictions that forecast higher than expected incidence are critical, as they could advocate increased interventions. To assess the performance of the forecast model framework beyond a simple seasonal profile, we defined a null model as the average DIR in each microregion for June 2000–2013. We consider the ability of both the forecast model and the null model to predict 'high risk' dengue across Brazil. *Table 2* shows a summary of contingency table results for observed DIR exceeding the high risk epidemic threshold (300 cases per 100,000

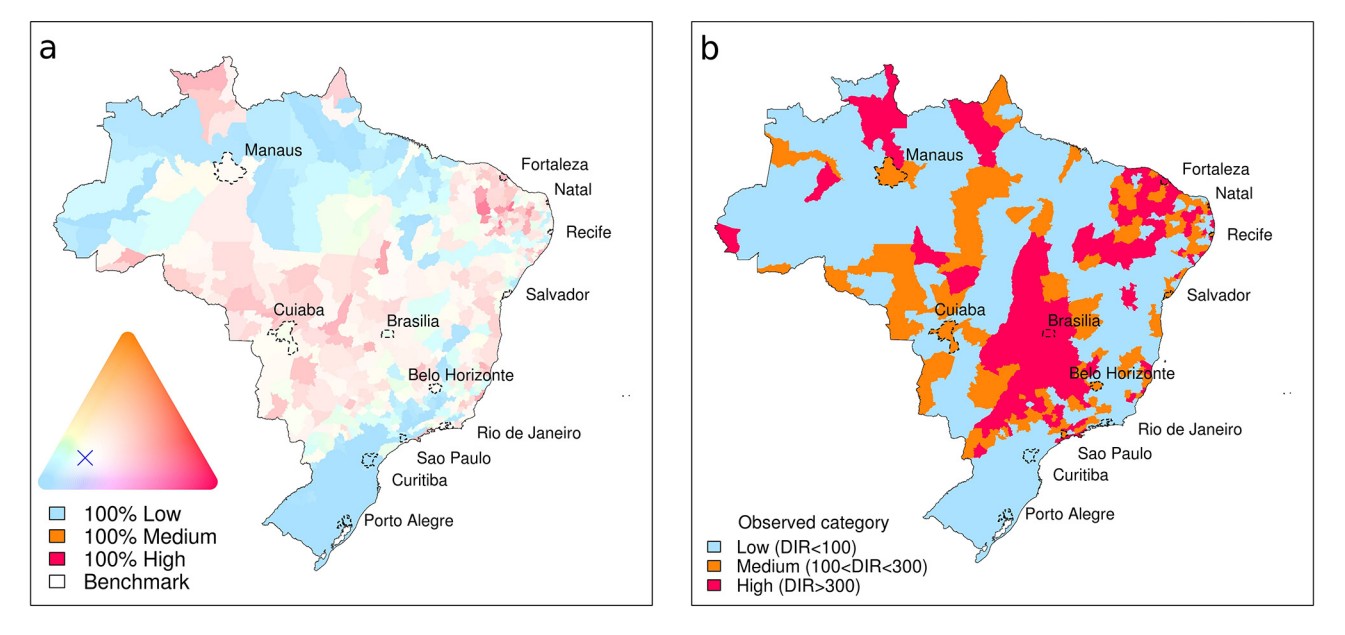

**Figure 2.** Probabilistic dengue forecast and observed dengue incidence rate categories for Brazil, June 2014. (**a**) Probabilistic dengue forecast for June 2014. The continuous colour palette (ternary phase diagram) conveys the probabilities assigned to low-risk, medium-risk, and high-risk dengue categories. Category boundaries defined as 100 cases per 100,000 inhabitants and 300 cases per 100,000 inhabitants. The greater the colour saturation, the more certain is the forecast of a particular outcome. Strong red shows a higher probability of high dengue risk. Strong blue indicates a higher probability of low dengue risk. Colours close to white indicate a forecast similar to the benchmark (long-term average distribution of dengue incidence in Brazil, June, 2000–2013: $p_L$=68%, $p_M$=16%, $p_H$=16%), marked by a cross. (**b**) Observed dengue incidence rate (DIR) categories for June, 2014.

inhabitants) using the probabilistic category forecast model and the null model for June 2014. Results show that for the June 2014 event, the forecast model predicted a greater number of true positives (hits) and fewer false negatives (misses) than the null model (see *Table 2*, Materials and methods). This gave a hit rate of 57% (miss rate of 43%) when using the forecast model and a hit rate of 33% (miss rate of 67%) when using the null model. However, the forecast model also tended to produce more false positives (or false alarms) than the null model (see *Table 2*). The two types of error (false alarms and missed events) have very different consequences for public health. For example, failing to predict an epidemic that then occurs (type II error – a miss) is much more damaging than predicting an epidemic that does not materialise (type I error – a false alarm) (*Stephenson, 2000*). *Figure 5* shows hit rates and false alarm rates for both the forecast and null model, calculated in 'leave one year out' cross-validation mode from 2000–2013, i.e. by excluding the year for which the prediction is valid when estimating model parameters (see Materials and methods). Results for the 2014 event are also included. The hit rate for the forecast model exceeds that of the null model for all years, expect 2004, when dengue incidence was at its lowest across the whole of Brazil.

To assess the additional value of the forecasting system beyond that of the seasonal profile, it is useful to consider the full posterior predictive distributions from the model, compared to the null model and associated prediction intervals. *Figure 6* shows time series of observed and predicted dengue incidence rates for June 2000–2014 for the 12 host microregions. The posterior predictive mean and upper 95% prediction (credible) interval from the forecast model, and the sample mean and upper 95% prediction interval from the null model are also included. The forecast and null model predictions are calculated in 'leave one year out' cross-validation mode from 2000–2013, i.e., by excluding the year for which the prediction is valid when estimating model parameters (see Materials and methods). Note, the predictions for 2014 are the only 'true' forecasts (i.e., no information is included beyond the forecast issue date). When considering the posterior predictive mean of the forecast model, some of the inter-annual variations in the observations are captured by the model, for example in Belo Horizonte, Manaus and Salvador. However, in some other places, the mean

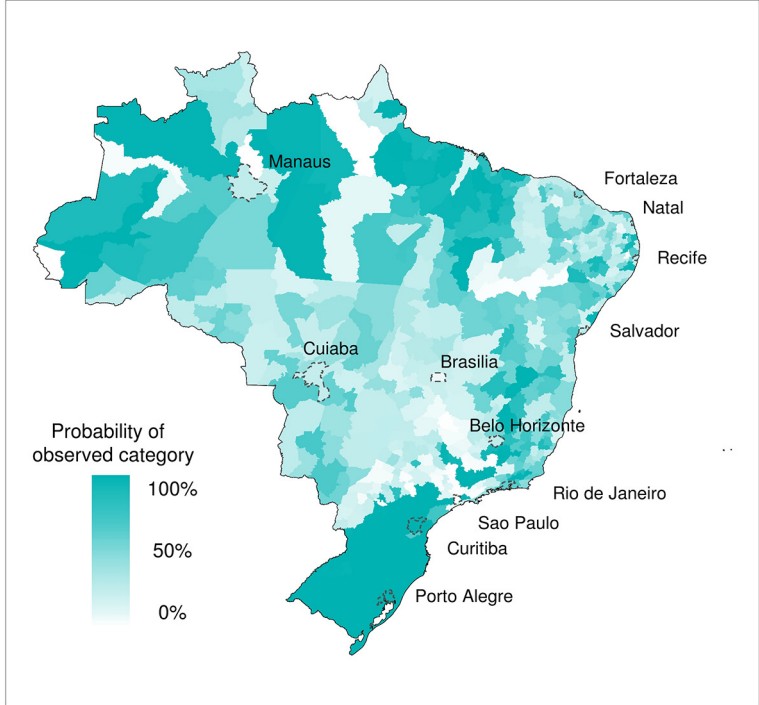

**Figure 3.** Forecast probability of observed DIR categories for June 2014. Probability of observing the correct DIR category (low, medium and high). The graduated colour bar represents the probability of observing any given category (ranging from 0%, pale colours, to 100%, deep colours).

prediction from the forecast and the null model are either very similar or, in some years, the null model mean is closer to the observed value, for example, in Cuiabá in 2009. During years with relatively low DIR, the predictions from the forecast model tend to be more precise than the null model, with narrower prediction (credible) intervals. Further, when DIR is exceptionally high, the forecast model is able to account for this increased possibility of an outbreak in most cases, compared to the

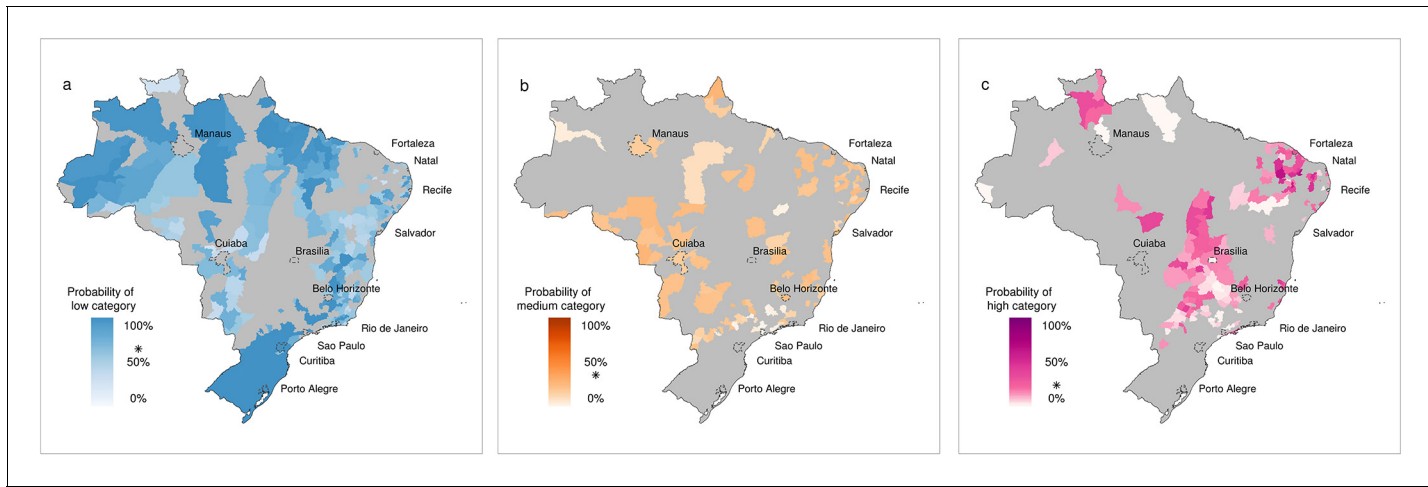

**Figure 4.** Forecast probability of observed DIR in the low, medium and high category for June 2014. Forecast probability given that (**a**) low, (**b**) medium and (**c**) high DIR was observed. Grey areas indicate that other DIR categories were observed and are therefore not considered. The graduated colour bar represents the probability of observing the given category (ranging from 0%, pale colours, to 100%, deep colours). Note, the alarm trigger thresholds are marked with a star (*). For (**a**) low risk warnings, $p_L > 68\%$, for (**b**) medium risk warnings, $p_M > 32\%$ and for (**c**) high risk warnings, $p_H > 18\%$. Colour bars are weighted, with increased saturation beyond the alarm trigger threshold to reflect the correct assignation of warnings.

**Table 2.** Summary of contingency table results for observed DIR exceeding the epidemic risk threshold. Summary of contingency table results for observed DIR exceeding the high risk epidemic threshold (300 cases per 100,000 inhabitants) using the probabilistic category forecast model and the null model (mean DIR, June 2000–2013) for June 2014.

| Performance measures | Forecast model probabilistic | Null model seasonal mean |
|---|---|---|
| Hit | 81 | 46 |
| False alarm (type I error) | 94 | 55 |
| Miss (type II error) | 60 | 95 |
| Correct rejection | 318 | 357 |
| Hit rate | 57% | 33% |
| False alarm rate | 23% | 13% |
| Miss rate | 43% | 67% |

null model. This is evident for the dengue epidemics that occurred in Belo Horizonte in 2010 and 2013 (*Figure 6a*), Salvador in 2010 (*Figure 6j*) and Manaus in 2011 (*Figure 6f*). Although the forecast model is far from perfect, in general, it is better able to detect extreme dengue incidence rates than the null model.

## Discussion

In this study, we have evaluated the ability of the dengue early warnings produced three months ahead of the 2014 World Cup to anticipate dengue risk categories, using pre-determined risk and alarm trigger thresholds. The forecasts correctly predicted dengue risk categories for seven of the twelve host microregions: Fortaleza and Natal (high), Belo Horizonte, Manaus and Salvador (medium) and Curitiba and Porto Alegre (low). The model was able to detect medium risk for Recife (posterior mean DIR=142, observed DIR=161, see *Figure 1*), although the definition of the alarm trigger thresholds placed this microregion in a higher risk category (the probability of high risk just exceeded 18%). The model missed the higher incidence that was observed in Brasília and São Paulo.

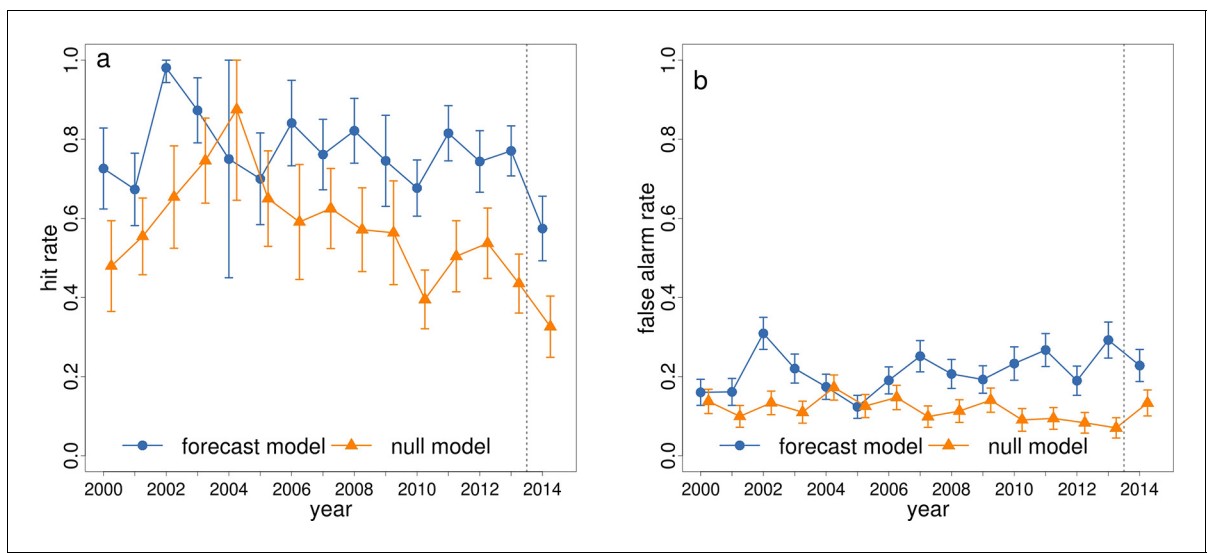

**Figure 5.** Hit rate and false alarm rate for predicting dengue in the high risk category for June 2000–2014 using the forecast model and null model. Comparison of (a) hit rates and (b) false alarm rates for the event of observed DIR exceeding the high risk epidemic threshold (300 cases per 100,000 inhabitants) using the probabilistic category forecast model (blue circles) and the null model (orange triangles) for June 2000–2014. The vertical bars around each point represent the 95% confidence intervals. The vertical dotted line separates the leave-one-out cross validation results (2000–2013) from the true predicted results for 2014.

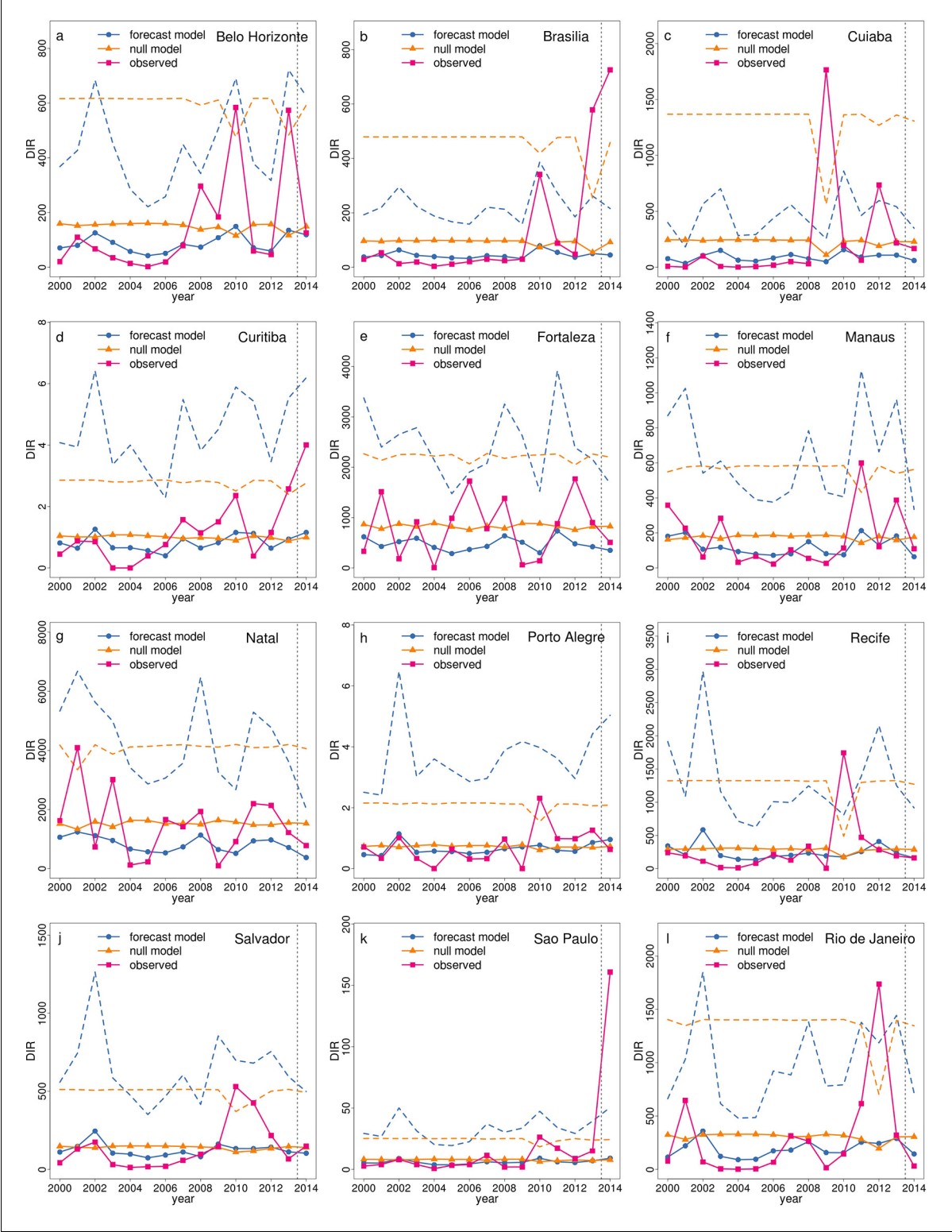

**Figure 6.** Time series of observed and predicted DIR for June 2000–2014 for host microregions. Observed DIR (pink squares), posterior mean DIR (blue circles) and upper 95% prediction (credible) interval from forecast model (blue dashed line) and mean DIR (orange triangles) and upper 95% prediction interval (orange dashed line) from null model, June 2000–2014 in the host microregions (**a**) Belo Horizonte, (**b**) Brasília, (**c**) Cuiabá, (**d**) Curitiba, (**e**) Fortaleza, (**f**) Manaus, (**g**) Natal, (**h**) Porto Alegre, (**i**) Recife, (**j**) Salvador, (**k**) São Paulo and (**l**) Rio de Janeiro. The vertical dotted line separates the leave-one-out cross validation results (2000–2013) from the true predicted results for 2014.

The observed dengue incidence rate for Rio de Janeiro was lower than usual for this time of year. When comparing the ability of the forecast model to the null model to predict high dengue risk across Brazil, the forecast model produced more hits and fewer missed events than the null model, with hit rates of 57% for the forecast model compared to 33% for the null model. The hit rate was almost always greater for the forecast model during previous years, 2000–2013. Despite the tendency of the forecast model to over predict dengue risk compared to the null model, we generally found the forecast model better able to detect extreme dengue incidence rates. For catastrophic events, such as epidemics, failing to predict an epidemic (e.g. forecast warning is for low risk but observed incidence is high) is considered much more damaging than announcing that an epidemic is about to occur and then does not happen (e.g. forecast warning is for high but observed incidence is low). Besides, if the dengue control programme intervene based on the forecast warnings, this could result in a decrease in cases compared to what would have happened otherwise.

The month of June is the transition in the southern hemisphere between autumn and winter, with the intensification of lower temperatures in central and southern Brazil (from Rio Grande do Sul to São Paulo and high plateaus of Minas Gerais), leading to areas with lower incidence rates. The model could, in general, distinguish climate-limiting areas for dengue fever (*Barcellos and Lowe, 2014b*). However, the model was unable to predict the high risk category for Brasília and São Paulo, which experienced unprecedented dengue incidence for the month of June. As the model is formulated using past data, and driven by climate forecasts and dengue data with a 3–4 month lead time, such irregularities and sudden shifts in transmission patterns can be difficult to capture. However, potential variability is accounted for in the random component of the model and correctly reflected in the probabilistic nature of our predictions.

The areas with greater model uncertainty (see pale areas in *Figure 2a*) are located along the fringes of climatic zones: the southern limits between the tropical warm and temperate climates, near the tropic of Capricorn; the Amazon rainforest contact with the tropical savanna vegetation (Cerrado); and the semiarid boundaries in the Northeast region. In fact, these fringes were the areas where there was a low probability of observing the correct category (see *Figure 3*). From the meteorological perspective, these zones should not be assumed as fixed boundaries, as they may vary over the years under the influence of the El Niño-Southern Oscillation, dominant air masses and other transient drivers. Important dengue outbreaks have been observed along these fringes that may be seen as a prelude of the expansion of dengue transmission area (*Barcellos and Lowe, 2014b*).

The dengue early warning model takes into account seasonal climate forecasts and epidemiological data for the months preceding the event, which create the conditions (environmental and epidemiological) for vector density and distribution, as well virus circulation. These conditions largely influence the development of an epidemic. Ideally, data about health care provision and vector control intensity between regions and over time would be included in the model. However, in the absence of such data, spatial and seasonal heterogeneity and dependency structures are accounted for in the model via random effects at the scale of calendar month and microregion, to try and ensure that the variance in the predictions includes uncertainty that could be generated by these unknown features of the disease system. Temporal random effects were not included in the model. Such effects are useful to help describe variations in temporal patterns and understand the relative contribution of various temporal explanatory variables. However, when it comes to predicting into the future, these effects become obsolete. In reality, dengue outbreaks are also modulated by factors that occur in more restricted temporal and spatial scales, which are not accounted for in the model framework, thus limiting the results of this study (*Lowe, 2015*). Such risk factors include the implementation and efficiency of vector control activities, circulating serotypes and herd immunity, along with other complex socio-economic factors, such as water storage practices.

From the start of 2014, a high pressure system over Southeast Brazil prevented the entrance of cold frontal systems from the south and the transport of humid air from the Amazonian Region in northern Brazil towards the southeast region of Brazil (*Coelho et al., 2015b*). This led to severe drought conditions in the South East region, which particularly affected the states of São Paulo and Minas Gerais. This situation led to a water supply system crises that began in the first semester of 2014 (*Coelho et al., 2015a*). This likely resulted in an increase of artificial water storage containers, increasing mosquito breeding sites and thus intensifying dengue transmission in this region (*Otto et al., 2015*). The combination of introducing new serotypes of dengue virus in a large city,

with a huge number of susceptible inhabitants, with high mobility and circulation of tourists, during a water crisis may have had an explosive effect on dengue transmission in this region. The unforeseen dengue outbreak in the microregion of Brasília may have resulted from a change in the dominant serotype to DENV1 (*Ministério da Saúde, 2014*) and the immunological status of the population. However, this requires further investigation. The four month lag between the target forecast date (June 2014) and the dengue cases observed before the forecast issue date (February 2014) was perhaps too long to capture these unprecedented changes in the dengue dynamics. In an ideal situation, the forecast would be updated each month leading up to the forecast target date, to incorporate the latest dengue and climate information available.

The objective of the model framework was to predict the probability of exceeding pre-defined 'epidemic' thresholds (see Materials and methods). The idea of a forecast that provides the probability of exceeding an epidemic threshold is to allow decision makers to quantify the level of certainty of the model predictions. For example, farmers often consult seasonal climate forecasts for their region of interest to decide when, how and what to plant their crops (*Meinke and Stone, 2005*). The information provided is given in a probabilistic context, e.g. "there is a 70% chance of observing temperatures above normal next season" rather than a specific value, e.g. "the temperature next season will be 27°C". Considering a dengue forecast, if the probability of exceeding the high dengue risk threshold for June in a given location is 90%, public health officials may be inclined to increase dengue control measures in that area (*Lowe et al., 2015a*). This information is likely to be more informative and beneficial to decision makers than a deterministic forecast indicating that "there will be a DIR of 182 in June".

The advantage of issuing probabilistic forecasts is that we can more easily quantify prediction uncertainty. However, effectively communicating probabilities to decision makers, the general public and even other scientists can be challenging. The use of set incidence (risk) thresholds and static probability decision (i.e. alarm trigger) thresholds means that the translation of probabilities into discrete warnings (low, medium and high) might not always reflect the predictive power of the model. For example, even though the mean estimate of the predictive distribution might be close to the observations (e.g. for the case of Recife), the predetermined cut-off alarm trigger thresholds might result in the designation of an incorrect categorical warning. Risk and alarm trigger thresholds need to be carefully chosen, as this influences the outcome of the 'translation' of probabilistic information into categorical risk warnings for decision makers.

The dengue incidence rate baseline has been increasing over the years. Since 1998, the overall incidence rate reached a plateau, situated between 100 and 300 cases per 100,000 inhabitants (*Barreto et al., 2011*). However, this seems to be a consequence of the spread of the disease in the country rather than an increase of incidence rates in cities (*Barcellos and Lowe, 2014b*). For example, the number of cities reporting dengue cases increased since the late 1980's from a few hundred to almost 4000, reaching 72% of the Brazilian territory (*Barreto et al., 2011*). The three class dengue risk levels (low<100, 100<medium<300 and high>300 cases per 100,000 inhabitants per year) used by the National Dengue Control Programme have been shown to appropriately capture the occurrence of large epidemics for certain cities, such as Rio de Janeiro (*Lowe et al., 2013*). For this city, the thresholds of 100 and 300 cases per 100,000 inhabitants has been adequate for distinguishing the recent outbreaks during the peak season (February-April) in the city since the year 2000. Using these thresholds, three large outbreaks in 2002, 2008 and 2012 (DIR>300), and smaller outbreaks in 2011 and 2013 (DIR>100) would have been detected (data not shown). However, these thresholds may not be appropriate for other cities, regions or seasons. This could be revised by taking into account population variability between microregions. For example, in the state of São Paulo, the decision to clinically or epidemiologically confirm dengue cases (see Materials and methods) is based on dengue incidence rates exceeding the high risk category threshold in small cities and the medium risk category threshold for large cities (*SES, 2010*). Recently, several different outbreak definitions have been tested using dengue data for Brazil, including moving averages, cumulative mean and fixed thresholds (*Brady et al., 2015*). However, there is still no clear recommendation as to the most appropriate threshold(s) to apply for a nationwide dengue early warning system in Brazil. The current dengue risk thresholds are defined using yearly dengue incidence rates and applied to the whole country. These could be refined by considering seasonal dengue variations in different regions across Brazil to create location specific thresholds, better able to detect the onset of dengue outbreaks at different times throughout the year.

The model framework could be improved by revising the definition of the alarm trigger thresholds, which are currently determined by assessing the ability of the model to distinguish medium and high risk categories in the past. These could be developed by stochastically simulating the optimum threshold for each category boundary multiple times, to produce a probability distribution of alarm trigger thresholds. Therefore, credible intervals could be assigned to the alarm trigger thresholds. This could help to flag warnings that might border two categories, e.g. low-medium or medium-high. For example, for Recife, the probability of high risk was 19% and the high risk alarm trigger threshold was a point estimate of 18%, which placed Recife into the high category, even though the predicted probability was just 1% greater than the trigger threshold. Ideally, a cost-benefit analysis should be performed by the public health services to assess the economic cost of false positives versus false negatives, to define optimum alert trigger thresholds. Another point to consider is the extent to which forecast warnings, issued several months in advance, influence vector control and personal protection activities. If a dengue forecast warning provokes more intense interventions or increased education and public awareness, this could potentially change the course of the dengue transmission patterns expected due to the environmental conditions and epidemiological trends.

Despite these limitations, the analyses performed in this study revealed several areas across Brazil with successful predictions. These results are promising for the future development of a routine dengue early warning system for Brazil. This will be conducted in consultation with the Brazilian Ministry of Health to align model capabilities with decision-maker requirements and data availability. In practice, early warning systems for such a complex disease should include several surveillance activities. Our model provides advance warning of the likely dengue risk three months later, based on climate forecasts and the current dengue risk situation at the time of forecast. This information should be updated and complimented with surveillance on the (re)introduction of new serotypes and notification tracking to ensure the ability of local health services to stay alert for dengue diagnosis. More widespread and better designed surveys to track changes in dengue seroprevalence should be routinely conducted. Establishing a new seroprevalence surveillance system at the scale of the whole country is a challenge. A national system could be initiated in key capital cities, to capture the entrance of new pathogens from other states or countries. The probability of introducing new serotypes could be determined by the number of people entering local airports and bus terminals from endemic regions. Given prior knowledge of seroprevalence, an estimate of the population at risk stratified by serotype could be incorporated in the model offset. By integrating data from the sentinel dengue seroprevalence surveillance systems at the time of forecast, predictions for each serotype could then be produced, to assess the epidemic potential generated by a re-emerging or dominant stereotype.

The model framework could also be downscaled and refined to account for within city disease dynamics and consider the impact of climate of other mosquito-transmitted viruses, such as chikungunya and Zika, which have recently emerged in Brazil (*Nunes et al., 2015*; *Zanluca et al., 2015*). This could help to assess the risk of arboviral disease importation and transmission in Rio de Janeiro ahead of the 2016 Olympics.

Several research groups developed dengue risk maps and forecasts ahead of the World Cup (*Barcellos and Lowe, 2014a*; *Hay, 2013*; *Lowe et al., 2014*; *Massad et al., 2014*; *van Panhuis et al., 2014*). While results varied between studies, all agreed that the northeast cities of Fortaleza and Natal would be at greater risk of dengue transmission and that the risk for the southern cities of Porto Alegre and Curitiba would be very low. *Massad et al. (2014)* also forecast a greater risk to tourists in the city of Rio de Janeiro. Only one study corrected predicted elevated dengue risk for teams and tourists in Brasília, using an Empirical Bayes model and weekly dengue incidence data up to May 2014 (i.e., one month lead time) (*van Panhuis et al., 2014*). Given the different research questions, methods, data and underlying populations that were used for each study, the direct comparison of these model results is not feasible in the absence of a consensus to translate results into equivalent quantities (*Lowe et al., 2015*). For example, the system evaluated here generated probabilistic distributions of yearly equivalent dengue incidence rates in the Brazilian population, at the monthly time scale and microregion spatial scale with 3–4 months lead-time, while other studies estimated the number of visitors at risk of dengue infection in specific cities. This emphasises the importance of dengue model inter-comparison projects, such as the 'Epidemic Prediction Initiative' launched by the White House Office of Science and Technology Policy (http://dengueforecasting.

noaa.gov/). The comparison of different models, forecasting common targets, will identify strengths and weakness in model formulation and gaps in data and methods.

## Conclusion

Several studies have developed models for dengue fever, using climate and other risk factors, and tested predictive performance in retrospective mode. However, none of these studies have incorporated real-time seasonal climate forecasts and epidemiological data to predict future dengue risk. Therefore, to our knowledge, this work constitutes the first evaluation of a nationwide dengue early warning, issued before a global mass gathering. The dengue early warnings were disseminated to the Ministry of Health, the general public and visitors travelling to Brazil, prior to the World Cup. The predictions were incorporated into the European Centre for Disease Control (ECDC) health risk assessment (*ECDC, 2014*) and reported by more than 18 international press outlets. As a result, the forecast by *Lowe et al. (2014)*, along with others (*Hay, 2013*; *Massad et al., 2014*), further contributed by raising general awareness about dengue fever and the risk of contracting the disease when travelling to endemic regions. This dengue early warning framework may be useful, not only ahead of mass gatherings, but also before the peak dengue season each year, to control or contain potentially explosive dengue epidemics. The use of real-time seasonal climate forecasts and early epidemiological reports in routine dengue early warnings is now a priority for the Brazilian Climate and Health Observatory (www.climasaude.icict.fiocruz.br), in collaboration with the Brazilian Institute for Space Research. We hope this prototype will serve as a demonstration for scientists, health surveillance teams and decision makers of the data and tools required to produce, communicate and evaluate timely predictions of climate-sensitive disease risk.

## Materials and methods

### Data

We obtained dengue data for June 2014 from the Notifiable Diseases Information System (SINAN), organised by the Brazilian Ministry of Health. We then aggregated the cases for the 5570 municipalities, 42% of which have less then 10,000 inhabitants, to the microregion level. A microregion is defined as an aggregate of neighbouring municipalities, with common economic interests and frequent population exchanges. This helps to alleviate problems of low population numbers and misreporting, due to variations in availability of health services/epidemiological facilities at the municipality level. This data includes confirmed cases of dengue fever, including mild infections, dengue haemorrhagic fever, and shock syndrome. Dengue cases can be confirmed by laboratory exams or clinical and epidemiological evidence. In the second case, a patient must present at least two of the following symptoms: high fever, severe headache, severe eye pain, joint and muscle pain, mild bleeding manifestation or low white cell count. In addition to these symptoms, the patient must have been in areas where dengue is being transmitted or where there has been an infestation of *Ae. aegypti* in the past 15 days (*Ministério da Saúde, 2005*). Dengue notification records are considered a priority in the epidemiological surveillance system in Brazil. Data flow is accelerated in relation to other diseases. About 50% of cases are reported within 3 days after the first symptoms, and 90% of cases are digitised within 7 days of notification (*Barbosa et al., 2015*). During outbreaks, this flow tends to be speeded up, adopting optimisation measures, for example, using 'clinical and epidemiological evidence' to confirm cases that could not be submitted for laboratory confirmation. These criteria allow accelerated case reporting, prioritising sensitivity and opportunity rather than specificity of information (*Duarte and França, 2006*). In fact, during high incidence periods, the proportion of cases confirmed by laboratory criteria is lower than during low incidence periods and 'clinical and epidemiological evidence' is a common procedure for case confirmation. On the other hand, about 50% of suspect cases are subsequently confirmed after laboratory tests in epidemic periods, while in periods of low transmission intensity, approximately 30% of cases are confirmed, revealing good predictive value of suspected cases (*Barbosa et al., 2015*). Therefore, after post-processing, some cases are discarded because they have negative serology. However, beyond these 30% of cases, identification of serotype is not carried out by laboratory tests, which hinders the understanding of transmission dynamics and population susceptibility level. Since recent data are still subject to confirmation of cases and elimination of duplicate registers, the initial figures of dengue cases may be

modified in the following months and will most likely be official by the end of 2015. These data will then be made publicly available via the Health Information Department (DATASUS, http://dtr2004. saude.gov.br/sinanweb/).

We used 2014 population estimates obtained from the Brazilian Institute for Geography and Statistics (*IBGE, 2014*) to convert the case data into dengue incidence rates (DIR), per 100,000 inhabitants. Other estimates of population for inter census years are produced by national institutions and may present discrepancies, mainly for small populations and newly created municipalities. According to the methodology used by the Ministry of Health, the DIR is calculated for a geographical space in a given year (*PAHO, 2008*). As the dengue risk thresholds used by the National Dengue Control Programme are based on yearly dengue incidence rates (*Ministério da Saúde, 2008*), it is necessary to use a proportion (1/12) of yearly population estimate as the denominator in the dengue incidence rate calculation, to make use of this at a monthly time scale. Therefore, we converted the monthly incidence to yearly equivalent incidence to make use of the risk thresholds of 100 and 300 cases per 100,000 per year. This is consistent with the metrics published in the Epidemiological Bulletins of the Ministry of Health. Our goal was to provide measures that can be easily interpreted by the Dengue Control Programme and translated into well understood risk levels (low, medium, high).

## Dengue forecast formulation and translation

A spatio-temporal Bayesian hierarchical model (*Lowe et al., 2011*, *2013*, *2014*) was formulated, using monthly dengue cases, from 2000 to 2013, for 553 Brazilian microregions as the response variable. Based on findings from previous studies (*Lowe et al., 2013*), the climate variables used to formulate the model were three-month average temperature (*Fan and Van den Dool, 2008*) and precipitation (*Adler et al., 2003*) anomalies (departures from the long-term average), over the three months preceding the dengue month of interest. This is equivalent to a two month lag when considering the mid-point of the three month average. Lags of 1–3 months are typically used when modelling dengue (*Lowe et al., 2015b*), to try and capture the impact of rainfall on mosquito breeding sites and the effect of temperature on the mosquito life cycle, although these relationships are still not well understood. Other explanatory variables included population density, altitude, and dengue relative risk (ratio of observed to expected cases) lagged by four months. Zone-specific seasonality was accounted for using autocorrelated annual cycles (i.e. by allowing each calendar month to depend on the previous month) for different Brazilian ecological zones (e.g. Amazon, Caatinga, Cerrado, Atlantic Pampa, Pantanal). Unknown confounding factors (e.g. health care and vector control disparities between microregions) and dependency structures (i.e., human mobility between neighbouring areas) were allowed for using area-specific unstructured and structured random effects (see *Lowe et al. (2014)* for further details).

To produce the forecast for June 2014, the model was driven by (1) real-time seasonal precipitation and temperature anomaly forecasts (*Coelho et al., 2006*), produced in mid-February by the Center for Weather Forecasting and Climate Research (CPTEC) (valid for the March-May [MAM] season) and (2) the observed epidemiological situation (ratio of observed to expected cases) for February, 2014, collated in March, 2014 by the Ministry of Health [see *Lowe et al. (2014)* for details]. Note, the precipitation seasonal forecasts used in this study were produced by CPTEC as part of EUROBRISA: A Euro-Brazilian Initiative for improving South American seasonal forecasts (http:eurobrisa.cptec.inpe.br).

Posterior predictive distributions were simulated for every microregion to determine the probability of dengue incidence rates exceeding predefined risk thresholds (see *Figure 1*). Probability forecasts ($p_L$, $p_M$, $p_H$) were issued for low (fewer than 100 dengue cases per 100,000 inhabitants), medium (between 100 and 300 dengue cases per 100,000 inhabitants) and high (more than 300 dengue cases per 100,000 inhabitants) risk. These results were presented using a visualisation technique (*Jupp et al., 2012*), where the forecast for each microregion was expressed as a colour determined by a combination of three probabilities, with colour saturation used to indicate certainty for a particular category (see *Figure 2a*). We then used a receiver operating characteristic (ROC) analysis of past forecasts and observations from 2000–2013 to define optimal probability thresholds for warnings. If the probability of low risk was less than 68%, a medium risk forecast warning was issued. If the probability of high risk was concurrently greater than 18%, the forecast warning was upgraded to high risk (see *Table 1*).

**Table 3.** The four possible outcomes for categorical forecasts of a binary event.

| | | Event observed | | |
|---|---|---|---|---|
| | | Yes | No | Total |
| Forecast warning issued | Yes | Hit (*a*) | False alarm (*b*) | *a+b* |
| | No | Miss (*c*) | Correct rejection (*d*) | *c+d* |
| | Total | *a+c* | *b+d* | *a+b+c+d=n* |

## Dengue forecast evaluation

After the event, we compared the published probabilistic predictions with observed DIR data for June 2014. We defined a null model as the seasonal average of past dengue incidence (i.e., mean DIR for June 2000–2013). We assessed the ability of the forecast model and the null model to determine the binary event of DIR exceeding 300 cases per 100,000 inhabitants (i.e. the high risk threshold) for n=553 microregions in Brazil (*Table 2*). *Table 3* shows the two ways for the forecast to be correct (either a hit or a correct rejection) and two ways for the forecast to be incorrect (either a false alarm or a miss). Cell count *a* is the number of events correctly forecast to occur, i.e. the number of hits; cell count *b* is the number of events incorrectly forecast to occur, i.e., the number of false alarms; cell count *c* is the number events incorrectly forecast not to occur, i.e., the number of misses; and cell count *d* is the number of event correctly forecast not to occur, i.e., the number of correct rejections (*Jolliffe and Stephenson, 2012*). We calculated performance measures, such as the hit rate; the proportion of events (i.e., epidemics) that were correctly predicted (*a/(a+c)*, also know as true positive rate or sensitivity) and the false alarm rate; the proportion of events that were predicted but did not occur (*b/(b+d)*, also know as false positive rate or 1-specificity). The false alarm rate can be interpreted as the rate of making a 'type I error', whereas the 'miss rate,' equal to one minus the hit rate, measures the rate of making a 'type II error' (*Stephenson, 2000*).

We compared time series of observed and predicted DIR from the forecast and null model for June 2000–2014 in the twelve host microregions. The mean of the posterior predictive distribution from the forecast model and the 95% prediction (credible) interval, obtained from the 2.5% and 97.5% percentiles of the distribution were calculated for June 2014, using the forecast model fitted to data from 2000–2013 (note, the lower 95% prediction interval from the forecast model was nearly always equal to zero and is therefore not shown in the figures). For the years 2000–2013, the model was fitted 14 times, excluding one year at a time when estimating model parameters, to produce 'cross validated' predictions to test against 'out-of-sample' data (i.e. the year for which the predictions are valid).

For the null model, the 95% prediction intervals for the sample mean was calculated as $\overline{y} \pm t_{n-1,\alpha/2} s \sqrt{1 + \frac{1}{n}}$, where $\overline{y}$ is the sample mean, $t_{n-1,\alpha/2}$ is the 100(1-α/2)$^{\text{th}}$ percentile of *T* Distribution, with n−1 degrees of freedom, and *s* is the standard error. For the 2014 null model prediction, the mean, standard error and 95% prediction intervals were calculated using past data for June 2000–2013 (n=14). For 2000–2013, the null model mean, standard error and 95% prediction intervals were calculated in cross-validated mode, by excluding one year at a time (n=13) (note, the lower 95% prediction interval for the null model was nearly always less than zero and is therefore not shown in the figures).

## Acknowledgements

We are grateful to the Brazilian Ministry of Health for providing the dengue data. The dynamical ensemble forecast data was kindly provided by the European Centre for Medium-Range Weather Forecasts (ECMWF) as part of the EUROBRISA license agreement.

## Additional information

### Funding

| Funder | Grant reference number | Author |
|---|---|---|
| Seventh Framework Programme | DENFREE project,FP7-HEALTH.2011.2.3.3-2; 282378 | Rachel Lowe Xavier Rodó |
| Seventh Framework Programme | EUPORIAS project, FP7-ENV.2012.6.1-1; 308291 | Rachel Lowe Xavier Rodó |
| Conselho Nacional de Desenvolvimento Científico e Tecnológico | Produtividade em Pesquisa - PQ - 2013, 306863/2013-8 | Caio AS Coelho |
| Seventh Framework Programme | SPECS project, FP7-ENV-2012-1; 308378 | Caio AS Coelho David B Stephenson |
| Financiadora de Estudos e Projetos | Brazilian Research Network on Global Climate Change, Rede Clima - FINEP, 01.13.0353-00 | Christovam Barcellos |
| Conselho Nacional de Desenvolvimento Científico e Tecnológico | Brazilian Observatory of Climate and Health, 552746/2011-8 | Christovam Barcellos |
| Conselho Nacional de Desenvolvimento Científico e Tecnológico | Produtividade em Pesquisa - PQ - 2013, 309692/2013-0 | Marilia Sá Carvalho |
| Fundação de Amparo à Pesquisa do Estado do Rio de Janeiro | E-23557/2014 | Marilia Sá Carvalho |
| Fundação de Amparo à Pesquisa do Estado de São Paulo | BEPE 2014/17676-0 | Rafael De Castro Catão |

The funders had no role in study design, data collection and interpretation, or the decision to submit the work for publication.

### Author contributions

RL, CASC, CB, MSC, TCB, DBS, XR, Conception and design, Analysis and interpretation of data, Drafting or revising the article; RDCC, Conception and design, Acquisition of data, Analysis and interpretation of data, Drafting or revising the article; GEC, WMR, Acquisition of data, Drafting or revising the article, Contributed unpublished essential data or reagents

### Author ORCIDs

Rachel Lowe, http://orcid.org/0000-0003-3939-7343
Christovam Barcellos, http://orcid.org/0000-0002-1161-2753

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
