## [Decision Letter]

Thank you for submitting your work entitled "Evaluating probabilistic dengue risk forecasts from a prototype early warning system for Brazil" for consideration by *eLife*. Your article has been reviewed by two peer reviewers, and the evaluation has been overseen by Simon Hay (Reviewing Editor) and Prabhat Jha (Senior Editor). One of the two reviewers, Jane Messina, has agreed to share her identity.

The reviewers have discussed the reviews with one another and the Reviewing editor has drafted this decision to help you prepare a revised submission.

Summary:

This aims to evaluate the predictive accuracy of a dengue forecasting model trained on data 2000-2013 for real-time prediction in the year 2014. The manuscript is well written, detailed and comprehensive. However, while I appreciate the various challenges of the field, my main concern is that, based on the data presented, the forecasts do not seem to perform very favorably and it is only really the inclusion of the near-miss category (when there are only 3 categories) that results in attractive headline figures. If based solely on direct classifications (7 world cup sites are predicted correctly and 5 incorrectly and only 54% of all areas are correctly identified). A forecasting system is only useful when it correctly recommends changes in activities that would have happened anyway. It would be therefore nice to see these predictions compared to simple seasonal averages (as a null model) and over more than one year.

Essential revisions:

1) Performance measured against a wider range of continuous performance metrics. Hit, miss and near miss is not a sufficiently high benchmark.

2) Performance above and beyond simple seasonal profile should be the null model.

3) Finally, a much more nuanced and objective appraisal of the success of this warning system is required.

These concerns are expanded below.

It would be good to include a comparison with a null model based on historical averages. Useful predictions from a forecasting system are only likely to be those that recommend changes from the activities that would otherwise have taken place anyway (which are typically based on the normal dengue season) – and beyond that predictions that forecast higher than expected incidence, as recommending a reduction in vector control efforts is politically challenging). What are the model predictive performance statistics in areas where the seasonal forecast is wrong?

I would suggest reconsidering the scoring system for hit, near-hit and miss. With just three categories and with two of them counting positively to performance statistics, even a random prediction would perform well, particularly if you excluded areas in the south and north that have highly predictable dynamics. Surely a predicted low but observed high is worse than a predicted high and observed low?

What is the difference between pre and post processed dengue cases reports?– it would be nice to see some numbers on this- how close do they correlate and is this different in an outbreak year where the number of reports mean surveillance personnel are overwhelmed and there is a backlog

I appreciate that the risk thresholds are defined by the ministry of health and that there is already an extensive discussion on the limitations this place on the analysis, but it might be nice to discuss the impact of changing incidence over time on these thresholds (the number of cases and presumably incidence has increased significantly since 2000, but so to presumably has treatment and control capacity – does this mean outbreaks should be more frequent now than before?)

The manuscript is slightly longer than it needs to be in all sections with some repetition of methods of the current or previous Lowe et al. paper. Reference these details out where possible.

---

## [Author Response]

*1) Performance measured against a wider range of continuous performance metrics. Hit, miss and near miss is not a sufficiently high benchmark.*

We are very grateful to the reviewers for identifying the potential weakness of including a 'near hit' category to calculate evaluation metrics. We have now removed this from the analysis and included further performance metrics, including:

– Maps to show the probability of observing the correct category overall and for each observed category, to highlight the value of providing probabilistic information and identify areas where the model was more certain of observing the correct category.

– An assessment of the ability of the forecast model to correctly predict high dengue incidence rates across Brazil. The hit rate (the relative number of times an event was forecast when it occurred) and the false alarm rate (the relative number of times the event was forecast when it did not occur) are compared between the forecast model (the probability of exceeding the high risk category threshold) and a null model (seasonal averages, based on historical data for June 2000-2013). These performance metrics are shown for the 2014 event and for previous years.

– Time series of observed and predicted dengue incidence rates for June 2000-2014 for the twelve host microregions. Posterior predictive mean and 95% prediction (credible) intervals from the forecast model and sample mean and 95% prediction intervals from the null model are compared. Note, the forecast and the null model predictions are calculated in 'leave one year out' cross-validation mode from 2000-2013, i.e. by excluding the year for which the prediction is valid when estimating model parameters.

*2) Performance above and beyond simple seasonal profile should be the null model.*

As described above, we have compared the ability of the forecast model (the probability of exceeding the high risk threshold) to the null model (seasonal averages, based on historical data for June 2000-2013) to distinguish high dengue risk across Brazil. We have also provided time series of observed and predicted dengue incidence rates (using both the forecast and the null model) for all years in host microregions, June 2000-2014.

*3) Finally, a much more nuanced and objective appraisal of the success of this warning system is required.*

Performance metrics are now reported and discussed for both the dengue forecast model and the null model, to gauge an idea of the additional skill and value gained by using the reported early warning system.

*These concerns are expanded below. It would be good to include a comparison with a null model based on historical averages. Useful predictions from a forecasting system are only likely to be those that recommend changes from the activities that would otherwise have taken place anyway (which are typically based on the normal dengue season) – and beyond that predictions that forecast higher than expected incidence, as recommending a reduction in vector control efforts is politically challenging). What are the model predictive performance statistics in areas where the seasonal forecast is wrong?*

Thank you for this comment. We have now compared the ability of the forecast model and null model (seasonal averages, based on historical data for June 2000-2013) to distinguish high risk dengue. We have also provided time series of observed and predicted dengue incidence rates (using both the forecast and the null model) for all years, June 2000-2014.

*I would suggest reconsidering the scoring system for hit, near-hit and miss. With just three categories and with two of them counting positively to performance statistics, even a random prediction would perform well, particularly if you excluded areas in the south and north that have highly predictable dynamics. Surely a predicted low but observed high is worse than a predicted high and observed low?*

We agree with the reviewers. We have removed the inclusion of a near hit category and have instead compared the ability of the forecast model and the null model to distinguish dengue incidence rates exceeding the high risk threshold. We also compare time series of observed and predicted dengue incidence rates for both the forecast model and the null model, to gauge an idea of the additional skill gained from the proposed early warning system. A discussion of the potential impact of false alarms (predicted high but observed low, i.e. type I error) and missed events (predicted low but observed high, i.e. type II error) is now included.

*What is the difference between pre and post processed dengue cases reports?– it would be nice to see some numbers on this- how close do they correlate and is this different in an outbreak year where the number of reports mean surveillance personnel are overwhelmed and there is a backlog*

The following text has been added to the data section of the Materials and methods:

“Dengue notification records are considered a priority in the epidemiological surveillance system in Brazil. […]However, beyond this 30% of cases, identification of serotype is not carried out by laboratory tests, which hinders the understanding of transmission dynamics and population susceptibility level.”

*I appreciate that the risk thresholds are defined by the ministry of health and that there is already an extensive discussion on the limitations this place on the analysis, but it might be nice to discuss the impact of changing incidence over time on these thresholds (the number of cases and presumably incidence has increased significantly since 2000, but so to presumably has treatment and control capacity – does this mean outbreaks should be more frequent now than before?)*

Thank you for this interesting question. Yes, the incidence rate baseline has been increasing over the years (see Figure 7). Since 1998, the overall incidence rate reached a plateau, situated between 100 and 300 cases per 100,000 inhabitants. However, this seems to be more a consequence of the spread of the disease in the country rather than an increase of incidence rates in cities. The number of cities reporting dengue cases increased since the late 1980's from a few hundred to almost 4000, i.e., 72% of the Brazilian territory (Barreto et al., 2011).

Author response image 1.Dengue incidence and number of municipalities reporting dengue cases in Brazil, 1985-2010 (Source: Barreto et al., 2011).**DOI:**
http://dx.doi.org/10.7554/eLife.11285.012

These processes have several consequences: 1) The emergence of large outbreaks in cities without prior transmission (Barcellos and Lowe, 2013); 2) An increase in the frequency of outbreaks with low amplitude in cities with favourable climatic and socioeconomic conditions (e.g., Fortaleza and Recife); 3) the occurrence of large outbreaks with low frequency in some large and favourable cities (e.g., Rio de Janeiro and Belo Horizonte). However, the frequency of outbreaks is regulated by population susceptibility to the viruses. As a result, the peaks tend to occur every 4 years; a necessary time frame for replacement of susceptible population or the arrival of a new type of virus.

We have added the following text to the Discussion to summarise this point:

“The dengue incidence rate baseline has been increasing over the years. […] For example, the number of cities reporting dengue cases increased since the late 1980's from a few hundred to almost 4000, reaching 72% of the Brazilian territory (Barreto et al., 2011).”

*The manuscript is slightly longer than it needs to be in all sections with some repetition of methods of the current or previous Lowe et al. paper. Reference these details out where possible.*

Repetition has been removed and details previously published have been referenced, where appropriate.